# ELASTIC: Numerical Reasoning with Adaptive Symbolic Compiler

**Jiaxin Zhang**
University of Strathclyde
16 Richmond Street, Glasgow, G1 1XQ
`jiaxin.zhang@strath.ac.uk`

**Yashar Moshfeghi**
University of Strathclyde
16 Richmond Street, Glasgow, G1 1XQ
`yashar.moshfeghi@strath.ac.uk`

## Abstract

Numerical reasoning over text is a challenging task of Artificial Intelligence (AI), requiring reading comprehension and numerical reasoning abilities. Previous approaches use numerical reasoning programs to represent the reasoning process. However, most works do not separate the generation of operators and operands, which are key components of a numerical reasoning program, thus limiting their ability to generate such programs for complicated tasks. In this paper, we introduce the numEricaL reASoning with adapTive symbolIc Compiler (ELASTIC) model, which is constituted of the RoBERTa as the Encoder and a Compiler with four modules: Reasoning Manager, Operator Generator, Operands Generator, and Memory Register. ELASTIC is robust when conducting complicated reasoning. Also, it is domain agnostic by supporting the expansion of diverse operators without caring about the number of operands it contains. Experiments show that ELASTIC achieves 68.96 and 65.21 of execution accuracy and program accuracy on the FinQA dataset and 83.00 program accuracy on the MathQA dataset, outperforming previous state-of-the-art models significantly.[1]

## 1 Introduction

Recently, Pre-trained Language Models (PLMs) [1, 2, 3, 4, 5] show astonishing performance over reading comprehension tasks like SQuAD [6]. However, PLMs fall short of numerical reasoning over text [7], which requires conducting numerical reasoning based on understanding the text. Hence, numerical reasoning over text is more challenging than reading comprehension [8] and attracts the interest of the AI community. Previous approaches adopt the sequence-to-sequence architecture to generate the sequential format of numerical reasoning programs (see (b) in Table 1) [9, 10]. However, the sequential format could produce invalid expressions such as "$3 - ((2)$" because of the wrong position of parentheses [11]. To avoid this, some methods convert the reasoning program to the binary tree, then use the tree-decoder to generate the pre/post-order traversal sequence (see (c) in Table 1) [12, 13, 14]. Alternatively, FinQANet [15] represents the reasoning program in a flattened format and generates the right parentheses forcibly after generating two consecutive operands. To increase the scalability, NeRd [7] introduces the symbolic operations and generates the reasoning program as the nested compositional format (see (e) in Table 1). Researchers also investigate to capture valuable information between entities and numbers to improve numerical reasoning ability. Some works use PLMs [8, 7, 16], while others, like Li et al. [17] and Ran et al. [18], adopt graph neural network to encode the text.

Currently, proposed approaches struggle with two significant problems. Firstly, they are vulnerable to complicated numerical reasoning problems. The complicated numerical reasoning problems usually

---

[1]ELASTIC code can be found at `https://github.com/NeuraSearch/NeurIPS-2022-Submission-3358`

36th Conference on Neural Information Processing Systems (NeurIPS 2022).

contain a long reasoning program, in which the types of operators are diverse, and the number of operands is dynamic. Since most works do not separate the generation of operators and operands, their performance is hindered by cascading errors when encountering complicated tasks. Secondly, previous works lack extensibility for the operators, which arises from either the flaw of the model architecture or the representation format of the program, making them hard to apply to different data domains.

Table 1: An Example (from MathQA [19] dataset) requires solving the problem by conducting numerical reasoning. The numerical reasoning program could be represented by four different formats: sequential format, tree-traverse format, flatten format [15], or nested format. $\#n$ refers to the executable result from the $n$th sub-program, and const_2 refers to the constant number 2.

| |
|---|
| **Problem**: A small table has a length of 12 inches and a breadth of b inches. Cubes are placed on the surface of the table so as to cover the entire surface. The maximum side of such cubes is found to be 4 inches. Also, a few such tables are arranged to form a square. The minimum length of side possible for such a square is 80 inches. What is the number for b? |
| **(a) Numerical Reasoning Program**: $b = \sqrt{(\frac{80}{4})^2 - 12^2}$ |
| **(b) Sequential Format**: 
 $\sqrt{((80 \div 4) \times (80 \div 4) - (12 \times 12))}$ |
| **(c) Pre-order Traverse Format**: 
 $\sqrt{}, -, \times, \div, 80, 4, \div, 80, 4, \times, 12, 12,$ none |
| **(d) Flattened Format**: 
 divide(80,4)\|power(12,const_2)\|power(#0,const_2)\|subtract(#2,#1)\|sqrt(#3) |
| **(e) Nested Format**: 
 sqrt(subtract(power(divide(80, 4), const_2), power(12, const_2))) |

Hence, we present the numEricaL reASoning with adapTive symbolIc Compiler (ELASTIC) model. ELASTIC separates the generation of operators and operands, allowing it to be less influenced by the cascading error from the complicated reasoning. Moreover, ELASTIC is adaptable to the number of operands following an operator, making it domain-agnostic to support diverse operators. Specifically, ELASTIC contains an Encoder part extracting the contextual representations of the passage and question and a Compiler part generating the numerical reasoning program. The Compiler consists of four modules: Reasoning Manager, Operator Generator, Operands Generator, and Memory Register. We conduct experiments on two challenging datasets: FinQA [15], and MathQA [19]. Since FinQA and MathQA are collected from different domains: annual financial reports and GRE/GMAT, ELASTIC demonstrates its adaptability by achieving state-of-the-art results on both datasets. Furthermore, our ablation studies investigate how the length of the numerical reasoning program influences the model's numerical reasoning ability, which shows that ELASTIC is less liable to being influenced by the cascading error. In addition, we introduce the maximum Memory Departing Distance (M-MDD), which measures how difficult for the mode to use the executable results from the previous sub-program. We use M-MDD to demonstrate the necessity of the Memory Register in ELASTIC. The contributions of our work are: (1) we present a numerical reasoning model ELASTIC with good adaptability and elasticity, which separates the generation of operators and operands. ELASTIC achieves state-of-the-art results on two challenging datasets: FinQA and MathQA; (2) we introduce the design of separate modules and Memory Register, making ELASTIC perform stably on complicated numerical reasoning problems; (3) the proposed ELASTIC is domain agnostic because it supports diverse operators.

## 2   Related Work

Making models to conduct numerical reasoning has attracted the AI community since the last century [20]. Previous research has investigated making the model do numerical reasoning over text by using statistical learning methods to find a similar equation pattern [21, 22, 23, 24]. Since deep learning has recently achieved great success in many tasks, Wang et al. [25] propose DNS, which is the first deep learning model for solving math word problems as far as we know. After their work, researchers

try to find a better way to represent the numerical reasoning program. For example, Wang et al. [26], and Wang et al. [27] use the expression tree to represent the reasoning program. Sun et al. [11] create tree-decoder GTS, which generates prefix traverse sequence of the tree. Chiang et al. [28], and Qin et al. [13] extract the semantic information from the question and passage texts and want to connect them with the reasoning steps. In addition, Li et al. [29], and Zhang et al. [30] introduce the graph encoder to capture the structural information or syntactic information to capture the relation between numbers and entities. Shen et al. [31] propose a unified model, which uses both sequential and graph as the encoder, then uses seq2seq and tree decoder to generate the reasoning program.

Furthermore, several datasets are proposed to evaluate the model's numerical reasoning ability, such as Math23K [32] and HWMP [13]. There are also more challenging datasets, like ASdiv [33], and MathQA [19]. At the same time, Dua et al. propose DROP [34], which requires more than arithmetic operations to conduct numerical reasoning. State-of-the-art models like NeRd [7] and NumNet [18] are introduced to solve the DROP dataset. More recently, a dataset called FinQA [15] has been proposed, which is constructed from the annual financial report.

Despite the considerable success achieved by these approaches, Patel et al. [35] argues that some state-of-the-art models, like GTS [11] and Graph2Tree [30], only learn the statistical relation instead of numerical reasoning ability. Unlike previous works, our model ELASTIC separates the generation of operators and operands, allowing it to conduct complicated numerical reasoning. Moreover, ELASTIC is adaptable to the number of operands following an operator, making it domain agnostic.

## 3 Approach

Figure 1 shows the architecture of our ELASTIC model. ELASTIC consists of an Encoder part encoding the question text and problem text into contextual vectors and a Compiler part producing the numerical reasoning programs. The Compiler part consists of four modules: Reasoning Manager, Operator Generator, Operands Generator, and Memory Register. The Reasoning Manager leverages other modules to produce the numerical reasoning program. Since a complete numerical reasoning program usually contains several sub-programs, the generation steps between operators and operands are interchangeable. To help the following sub-programs use executable results from the previous sub-programs, Memory Register stores the sub-programs executable results into corresponding pre-defined cache tokens embeddings.[2]

Table 2: Task Definition Notation

| Notation | Description |
|---|---|
| $P, Q, R$ | Problem Text, Question Text, Numerical Reasoning Program |
| *NUM* | The numbers in $P$ and $Q$ |
| *CONS* | Constants defined in DSL |
| *OP* | All mathematical operators |
| $op_i$ | The $i$th operator in $R$ |
| *OE* | All operands |
| $oe^i$ | All operands belonging to $op_i$ |
| $oe^i_j$ | The $j$th operands of $op_i$ |
| $s$ | From either *OP* or *OE*, $s$ constitute $R$ |
| $r_i$ | The i-th sub-program of $R$ 
 $r_i = op_i \left[ oe^i \right]$ |

**Task Definition**    Given the problem text $P$ and question text $Q$, the task is to generate a numerical reasoning program $R$. Both problem text $P$ and question text $Q$ consist of words and numbers (denoted by *NUM*). The Numerical reasoning program $R$ represents the numerical reasoning process,

---

[2]See Appendix F for an example showing how different modules work.

which is a sequence of symbols (denoted by $s$) from mathematical operators (denoted by $OP$) and operands (denoted by $OE$). Operands $OE$ are from either constant numbers (denoted by $CONS$) defined in Domain Specific Language (DSL) or $NUM$. $CONS$ are the special numbers that do not exist in either the problem text $P$ and question text $Q$, such as const_pi($\pi$). Finally, the pattern of the numerical reasoning program $R$ is defined as $R = \left\{ op_i \left[ oe_j^i \right]_{j=0}^{m-1} \right\}_{i=0}^{n-1}$, where $op_i \in OP$, it is the $i_{th}$ operator in $R$, and $op_i$ contains several operands $oe_j^i$. In addition, we regard a group of one operator and its operands as the sub-program $r$. For example, $op_i \left[ oe_j^i \right]$ is the $i_{th}$ sub-program $r_i$, which can be executed since it is a complete arithmetic program.[3]

## 3.1 Encoder Part

As shown in Figure 1 (Encoder), the Encoder takes the concatenated sequence of $Q$ and $P$ as input. The Encoder encodes the input sequence and outputs the contextual vectors $\mathbf{h}^{enc}$. Next, $\mathbf{h}^{enc}$ is used for the Compiler to produce the numerical reasoning program $R$. In this work, we use RoBERTa as the Encoder. The outputs from the final layer of RoBERTa is used as $\mathbf{h}^{enc} \in \mathbb{R}^{h*s}$, where $s$ is the maximum input length of the RoBERTa, and $h$ is the hidden size of RoBERTa. Note that ELASTIC is not dependent on the specific type of encoder. Any model providing contextual vectors of the sequence can be used.

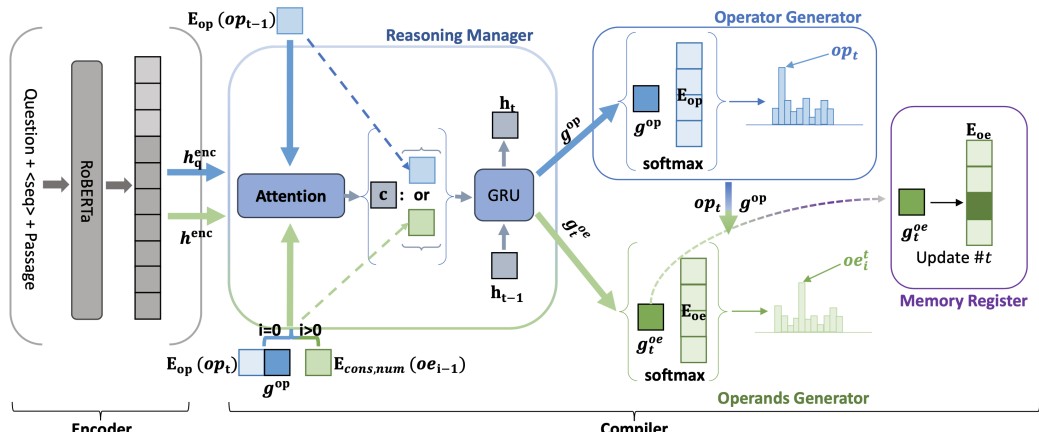

Figure 1: The overall architecture of the ELASTIC model. The Encoder part takes the sequence of question text $Q$ and passage text $P$ as input, then generates the contextual vectors $\mathbf{h}^{enc}$. The Compiler part consists of four modules: **Reasoning Manager**, **Operator Generator**, **Operands Generator**, and **Memory Register**. The right part of the figure shows a complete process of the generation of sub-program $r_t$. Firstly, Reasoning Manager sends the guidance vectors $g^{op}$ to the Operator Generator, which guides the generation of operator $op_t$. Secondly, Reasoning Manager suspends the Operator Generator, then the Operands Generator takes $g^{op}$ and $op_t$ from the Operator Generator to produce the first operand $oe_1^t$. When finish the generation of the sub-program $r_t$, the Memory Register stores the results and updates the embedding vectors of cache token #$t$ by $g_t^{oe}$. Again, the Compiler repeats to generate next sub-program $r_{t+1}$.

## 3.2 Compiler Part

**Decoding Vocabulary and Token Embedding**    We first describe the decoding vocabulary. The decoding vocabulary consists of $OP$ and $OE$, where $OE$ can be further categorized into $NUM$ and $CONS$. The embedding $\mathbf{e}_s$ of symbol $s$ of the decoding vocabulary is represented by the embedding $\mathbf{E}_{op,cons,num}(s)$, which is the embedding look-up function. Hence, the embedding for symbol $s$ is defined as:

---

[3]See Table 2 for the definition of all the notations. Also, see Appendix E for an example.

$$\mathbf{e}_s = \begin{cases} \mathbf{E}_{op}(\text{s}) & \text{if s} \in OP \\ \mathbf{E}_{cons}(\text{s}) & \text{if s} \in CONS \\ \mathbf{E}_{num}(\text{s}) = \mathbf{h}_i^{enc} & \text{if s} \in NUM \end{cases} \tag{1}$$

The symbols embeddings of *OP* and *CONS* are two trainable embedding matrices $\mathbf{E}_{op} \in \mathbb{R}^{h*n_{op}}$ and $\mathbf{E}_{cons} \in \mathbb{R}^{h*n_{cons}}$ ($n_{op}$ and $n_{cons}$ refers sizes of *OP* and *CONS* respectively). The embedding for the symbol of *NUM* is $\mathbf{h}_i^{enc} \in \mathbb{R}^h$, where $i$ denotes the index position in the sequence of $\mathbf{Q}$ and $\mathbf{P}$.

**Reasoning Manager** As shown in Figure 1 (Reasoning Manager), the Reasoning Manager outputs the vector $\mathbf{g}$, which guides the Operator Generator and the Operands Generator to produce *op* and *oe*. The inputs for the Reasoning Manager are contextual vectors $\mathbf{h}^{enc}$ ($\mathbf{h}_q^{enc}$ for generating operators) from the Encoder and embedding of the previously generated symbol $s_{t-1}$. The Reasoning Manager first calculates the context vector $\mathbf{c}$ by the normalized vectors of $\mathbf{h}_i^{enc}$ and the attention weights $a_i$:

$$\mathbf{c} = \sum_i a_i \mathbf{h}_i^{enc} \tag{2}$$

$$a_i = \frac{\exp(\text{score}(\mathbf{e}_{s_{t-1}}, \mathbf{h}_i^{enc}))}{\sum_j \exp(\text{score}(\mathbf{e}_{s_{t-1}}, \mathbf{h}_j^{enc}))} \tag{3}$$

$$\text{score}(\mathbf{e}_{s_{t-1}}, \mathbf{h}_i^{enc}) = \mathbf{e}_{s_{t-1}}^{\text{T}} \mathbf{W}_1 \cdot \mathbf{W}_2 \mathbf{h}_i^{enc} \tag{4}$$

where $\mathbf{W}_1 \in \mathbb{R}^{h*h}$ and $\mathbf{W}_2 \in \mathbb{R}^{h*h}$, and both are trainable parameters. The $\mathbf{c}$ summarizes the encoded information from the Encoder according to the previous generated symbol $s$. Next, the Reasoning Manager adopts the GRU [36] network to generate the guidance output $\mathbf{g}$:

$$\mathbf{g}, \mathbf{h}_t = \text{GRU}(\text{Relu}(\mathbf{W}_3[\mathbf{c} : \mathbf{E}_{op,cons,num}(s_{t-1})]), \mathbf{h}_{t-1}) \tag{5}$$

where ":" represents concatenation. $\mathbf{W}_3 \in \mathbb{R}^{h*2h}$ is trainable parameter, and Relu is the activation function. $\mathbf{h}_{t-1} \in \mathbb{R}^h$ is the hidden state of GRU from the previous step, and $\mathbf{h}_0$ is $\mathbf{0}$.

**Operator Generator** As shown in Figure 1 (Operator Generator). Firstly, the Operator Generator receives the guidance vector $\mathbf{g}_t^{op}$ from the Reasoning Manager by inputting: contextual vectors $\mathbf{h}_q^{enc}$ of tokens from the question $Q$, and embedding $\mathbf{E}_{op}(op_{t-1})$ of the previously generated operator. Next, the Operator Generator calculates the probabilities of $i$-th operator (denoted as $i$-$op$) of the OP:

$$\mathbb{P}(i\text{-}op|\mathbf{E}_{op}(op_{t-1}), \mathbf{g}_t^{op}) = \frac{\exp(\mathbf{E}_{op}^{\text{T}}(i\text{-}op)\text{Relu}(\mathbf{W}_{op}\mathbf{g}_t^{op}))}{\sum_{j\text{-}op\in OP} \exp(\mathbf{E}_{op}^{\text{T}}(j\text{-}op)\text{Relu}(\mathbf{W}_{op}\mathbf{g}_t^{op}))} \tag{6}$$

where $W_{op} \in \mathbb{R}^{h*h}$ is trainable parameter. The Operator Generator selects the operator with the highest probability as the predicted *op*. Next, unlike other models, the Reasoning Manager suspends the generation of operators and starts to generate operands $\{oe^t\}$ through the Operands Generator.

**Operands Generator** As shown in Figure 1 (Operands Generator). The inputs from Operands Generator to the Reasoning Manager are different from Operator Generator's. Because *oe* could be a number in $Q$ or $P$, the contextual vectors $\mathbf{h}^{enc}$ of all tokens are used. Furthermore, the Operands Generator initializes the embedding of the initial operand $\mathbf{e}_{(oe_0^t)}$ as $\text{Relu}(\mathbf{W}_4[\mathbf{E}_{op}(op_t) : \mathbf{g}_t])$ ($\mathbf{W}_4 \in \mathbb{R}^{h*2h}$), leveraging information of $op_t$ to produce $oe^t$. Next, the Reasoning Manager outputs $g_n^{oe}$ for $n$-th step generation of operand $oe_n^t$. Finally, the probability of $i$-th operand (denoted as $i$-$oe$) of the OE:

$$\mathbb{P}(i\text{-}oe|\mathbf{E}_{cons,num}(oe_{n-1}^t), \mathbf{g}_t^{oe}) = \frac{\exp(\mathbf{E}_{cons,num}^{\text{T}}(i\text{-}oe)\text{Relu}(\mathbf{W}_{oe}\mathbf{g}_t^{oe}))}{\sum_{j\text{-}oe\in OE} \exp(\mathbf{E}_{cons,num}^{\text{T}}(j\text{-}oe)\text{Relu}(\mathbf{W}_{oe}\mathbf{g}_t^{oe}))} \tag{7}$$

where $W_{oe} \in \mathbb{R}^{h*h}$ is trainable parameter. The Operands Generator selects the operand with the highest probability as the predicted $oe$. After one operand has been generated, the Operands Generator continues producing operands for the sub-program $r_t$. The decoding process for the operands terminates when the token *none* is produced.

**Memory Register**   When generating sub-program $r_i$, its operands could be the executable results from the previous sub-program $r_p(p < i)$. To make the Operands Generator be able to use the results from previous sub-programs. Inspired by Chen et al. [15], we introduce a cache token $\#n$ to the *CONS* of DSL, which is used for storing the information of executable results. Unlike other constants, $\#n$ does not point to a static value. It is different according to the different sub-program $r_n$. As the results, ELASTIC needs to update the representation of $\#n$ after the sub-program $r_n$ is generated. Specifically, the Memory and Register module update the cache $\#n$ by replacing its embedding with output $\mathbf{g}_n^{oe}$, which is the guidance vector from Reasoning Manager to guide the generation of the last operands belonging to the sub-program $r_n$.

**Training Objective**   Given the data $\mathbb{D}$ with size of $N$ containing $P_i, Q_i, \hat{op}^i, \hat{oe}^i$, where $P_i$ and $Q_i$ refer to the passage and question in the the $i^{th}$ training data, likewise, $\hat{op}^i$ and $\hat{oe}^i$ are the golden operators and operands. Our training goal is to minimize the sum of the negative log-likelihood over the entire data, so the training loss is $\sum_{i=1}^{N} - \left\{ \log \mathbb{P}(\text{OP}^i | \mathbf{P}^i, \mathbf{Q}^i) + \log \mathbb{P}(\text{OE}^i | \mathbf{P}^i, \mathbf{Q}^i) \right\}$ .

## 4   Experimental Set-up

**Datasets**   We conduct evaluation experiments on two datasets: FinQA [15] and MathQA [19].[4]

**FinQA:** FinQA is a dataset created from the annual financial reports. It contains 8,281 data, split into train, eval, and test parts with 6,251, 883, and 1,147 examples. We adopt the evaluation metrics from the original FinQA paper: execution accuracy (Exe Acc) and program accuracy (Prog Acc). The program accuracy calculates the accuracy of the operators and operands between the predicted program and the golden program. The execution accuracy calculates the accuracy between the golden executable result and the result from the predicted program. Since the FinQA dataset only contains operators with two operands, we extend it by creating questions required to be solved by the operators with more than two operands. We use the extended FinQA dataset to evaluate our models' adaptability to the number of operands (See Appendix A).

**MathQA:** MathQA is created from GRE/GMAT examinations, containing 37,200 math word problems. The dataset is split into $80\%$, $12\%$, and $8\%$ of train, dev, and test data. Compared with the FinQA dataset, the examples of MathQA require more advanced reasoning ability, which challenges the model to conduct advanced numerical reasoning (see Appendix B). A significant difference with FinQA is that the number of operands following an operator is not explicit in the MathQA dataset. Each MathQA question contains one correct of several answer options, calculated by the reasoning program with the knowledge of the operation semantics. Since we do not have this kind of knowledge, we adopted the same way as NeRd [7], by only using program accuracy to evaluate models' performances. Note that program accuracy is stricter than execution accuracy because the model could find the correct answer by spurious reasoning programs.

**Baselines**   We compare our ELASTIC model with several state-of-the-art models. (1) **FinQANet** [15]: It adopts the Encoder-Decoder architecture with a cache updating mechanism to generate the program. Since FinQANet only supports generating operators with exact two operands, we manage to train and evaluate FinQANet on the MathQA dataset by discarding the operators containing more than two operands. (2) **NeRd** [7]: it uses the BERT and a pointer-generator-based model to generate the symbolic nested program. (3) **Graph2Tree** [17]: It models the dependency information of the text sequence by the GraphSAGE [38] like model, and generates the program in a tree-structured way. (4) **NumNet** [18]: NumNet models the numeracy information by a GNN network. We also train the **NumNet+**, which replaces the Encoder of the NumNet by RoBERTa-large.[5] Note that program

---

[4]We do not select other datasets because of: (1) too small in size (around 1000), e.g., MAWPS [37] and ASDiv-a [33], (2) language are not English. e.g. Math23K [10] and HMWP [13], (3) lack intermediate annotated program, like DROP [34].

[5]`https://github.com/llamazing/numnet_plus`

accuracy does not apply to NumNet, since NumNet does not generate compositional reasoning programs. (5) **Human Performance**: We also report the human performance of both experts and non-experts in the FinQA dataset. The results are taken from the original FinQA paper [15].

**Implementation Details**    The model is implemented by Pytorch [39] and Transformer [40], then trained on a server with an NVIDIA Tesla A100 GPU of 40G memory. Training epochs are set to 50 and 100 for FinQA and MathQA, respectively. The batch size for all datasets is set to 10. We use Adam as optimizer [41] to update the parameters of the models. The initial learning rate is set to 1e-5 equally, and it would be halved in every 25 epochs and 50 epochs for FinQA and MathQA. During training, the dropout rate and the weight decay are set to 0.1 and 1e-5 to prevent over-fitting. The parameters of the RoBERTa are fine-tuned during training. For the GRU cell in the decoder, the hidden size is the same as the RoBERTa, and the GRU layers number is 4. During inference, we use greedy decoding to generate the reasoning program.

## 5    Results

**Overall Results**    Table 3 shows the performances of our ELASTIC model and baselines on FinQA and MathQA. Overall, ELASTIC (RoBERTa-large) achieves the highest scores on both datasets. In the FinQA dataset, we see a significant lead in our ELASTIC (RoBERTa-large) model compared to the best baseline FinQANet (RoBERTa-large), with 3.91 points higher execution accuracy and 1.69 points higher program accuracy. When we change the Encoder part of ELASTIC from RoBERTa-large to RoBERTa-base, it still achieves better results than FinQANet using the same size of RoBERTa. Since both ELASTIC and FinQANet use the RoBERTa as the encoder, the results demonstrate the improvements brought by separating the generation procedures for operators and operands. Both ELASTIC models outperform the NeRd with a large margin. It is worth mentioning that NeRd defines external rules for different operators in their model [7], which is not the case with the ELASTIC. ELASTIC also outperforms the NumNet and NumNet+ by a considerable margin. This could be due to the internal structure of these models limiting their scalability in generating reasoning programs, thus struggling to produce reasoning steps in a systematic manner [42]. Finally, Graph2Tree achieves only 0.37 accuracy on the FinQA test dataset, which is much lower compared to its 69.96 program accuracy on the MathQA dataset. We suspect that this is because of the data leak problem existing in FinQA train and eval data (see detailed explanation in Appendix D). Although ELASTIC surpasses the non-expert performance, we can still find a large gap between our ELASTIC model and Human Expert.

For the MathQA dataset, ELASTIC (RoBERTa-large) is the best performing model, with 3.3 higher program accuracy than NeRd and 13.04 points higher than Graph2Tree. We further investigate the performance of ELASTIC using RoBERTa-base, which still achieves higher accuracy of 82.27 than 79.7 of NeRd. The slight performance difference between ELASTIC (RoBERTa-large) and ELASTIC (RoBERTa-base) suggests that the extracted contextual semantic information of passage and question is sufficient. Finally, FinQANet achieves promising results on MathQA, 79.20% program accuracy by FinQANet (RoBERTa-large) and 74.12% program accuracy by FinQANet (RoBERTa-base). Note that we discarded the data of MathQA containing more than two operands for the FinQANet, so that performance of FinQANet on MathQA is the overestimation of its numerical reasoning ability. Even with such consideration, ELASTIC outperforms FinQANet significantly, demonstrating that ELASTIC is more adaptable by supporting diverse operators than FinQANet.

**Performance Breakdown**    To demonstrate the strength of ELASTIC, we investigate the importance of the Memory Register. Also, we show ELASTIC performance when generating different lengths of program steps.

**Necessity of Memory Register**    As discussed in the Section Memory Register, ELASTIC stores the executable results of each sub-program into a special cache token #n, and updates its embedding after $n$-th sub-program is generated. The longer the reasoning program is, the higher the probability of the generating process using the previous sub-program result. This section investigates the effect of the Memory Register on improving numerical reasoning performance.

First, we present an ablation study of using and not using the Memory Register for ELASTIC. From Table 4, we find that ELASTIC with Memory Register performs slightly better than it without. Similar

Table 3: Overall Results for the baselines and ELASTIC on the testing data from three datasets. † means that the scores are taken from the original papers. ‡ means that the scores are taken from the FinQA paper [15]. ⋆ The program accuracy does not apply to the NumNet on FinQA and MathQA datasets because NumNet does not generate the intermediate reasoning program. In addition, NumNet could only solve reasoning program involving add and subtract operations. However, the proportions of examples only use add and subtract as operations in MahtQA are $0.055\%$ and $0.056\%$, respectively. As a result, we choose not to train NumNet on MathQA.

| Datasets & Metrics | FinQA (test) | | MathQA (test) |
|---|---|---|---|
| | Exe Acc | Prog Acc | Prog Acc |
| Graph2Tree | 0.37 | 0.0 | 69.96† |
| NumNet | 2.32 | n/a⋆ | n/a⋆ |
| NumNet+ | 10.29 | n/a⋆ | n/a⋆ |
| NeRd | 52.48‡ | 49.90‡ | 79.70† |
| FinQANet (RoBERTa-base) | 60.10† | 58.38† | 74.12 |
| FinQANet (RoBERTa-large) | 65.05† | 63.52† | 79.20 |
| ELASTIC (RoBERTa-base) | 62.66 | 59.28 | 82.27 |
| ELASTIC (RoBERTa-large) | **68.96** | **65.21** | **83.00** |
| Human Expert | 91.16† | 87.49† | n/a |
| Human Non-Expert | 50.68† | 48.17† | n/a |

Table 4: The performances of ELASTIC with or without memory register (MR). ELASTIC with MR performs better than without MR on FinQA and MathQA datasets. Both ELASTIC with or without MR performs better than FinQANet. All models use the RoBERTa-large as the encoder.

| Datasets & Metrics | FinQA (test) | | MathQA (test) |
|---|---|---|---|
| | Exe Acc | Prog Acc | Prog Acc |
| ELASTIC *w* MR | **68.96** | **65.21** | **83.00** |
| ELASTIC *w/o* MR | 68.79 | 64.78 | 82.68 |
| FinQANet | 65.06 | 63.52 | 79.20 |

observations can be found for the MathQA dataset. This observation demonstrates the value of the Memory Register. Next, since ELASTIC and FinQANet store the executable results from the previous sub-program in a different way, we conduct a comparison between the two models. The results from Table 4 show that the ELASTIC with Memory Register achieves significantly higher scores than FinQANet on both datasets.

Next, given two sub-program belonging to the same $R$: $r_i$, $r_j$ ($i < j$). where the executable result of $r_i$ is used as the operand for $r_j$. Then, we introduce the Memory Departing Distance (MDD) for $r_i$ and $r_j$ as $j - i$, and the maximum Memory Departing Distance (M-MDD) as the longest MDD between all $\{r\}$ of $R$.[6] The bigger M-MDD is, the more challenging to select the correct previous sub-program result, since the model tends to forget the information passing from long steps before. As the result, we investigate how models perform when dealing with different M-MDD.

From Figure 2a and Figure 2b, the ELASTIC with Memory Register performs better than ELASTIC without it at each M-MDD on FinQA and MathQA datasets. Particularly in the MathQA dataset, when M-MDD is larger than 5, ELASTIC with Memory Register can achieve better results than the ELASTIC without it. This demonstrates the importance of the Memory Register when using

---

[6]For example, in flatten program "$\mathrm{add}(20, 3), \mathrm{subtract}(6, 1), \mathrm{add}(\#1, 10), \mathrm{subtract}(\#0, \#2)$", the MDD for $r_0$, $r_1$, and $r_2$ are 3, 1, and 1. Obviously, the maximum M-MDD is 3.

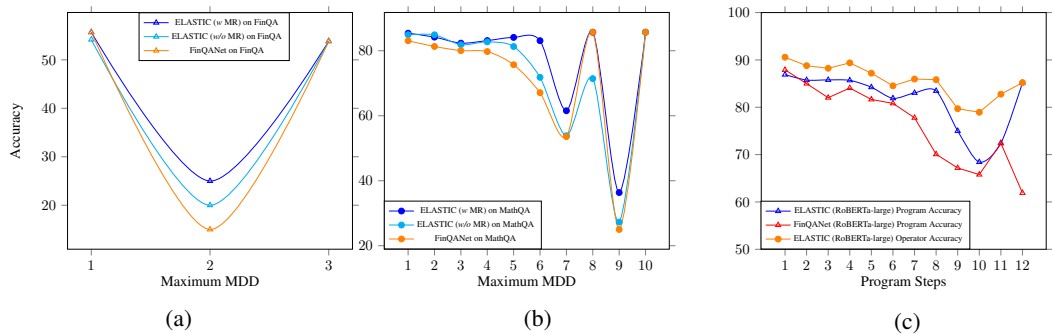

Figure 2: (a) Program Accuracy on FinQA according to the M-MDD. (b) Program Accuracy on MathQA according to the M-MDD. (c) Program Accuracy and Operator Accuracy of ELASTIC (RoBERTa-large) on different program steps in MathQA dataset, compared with Program Accuracy of FinQANet (RoBERTa-large).

executable results from long steps before. Worth mentioning that ELASTIC performs better than FinQANet on both datasets, even without the Memory Register.

**Performance on Different Program Steps**    When producing the long numerical reasoning program, ELASTIC is less influenced by the cascading error. To demonstrate this superiority of ELASTIC, we investigate how different lengths of programs influence models' performances.

Table 5: ELASTIC and FinQANet performances on FinQA dataset in terms of different program steps. The "# Train & Dev" is the number of training and development data. All models use the RoBERTa-large as the encoder. † means the results of that model are taken from the original FinQA [15] paper. ‡ means that the FinQANet (RoBERTa-large) is re-trained by ourselves.

| Program Steps | ELASTIC | | FinQANet† | | FinQANet‡ | | # Train & Dev |
|---|---|---|---|---|---|---|---|
| | Exe Acc | Prog Acc | Exe Acc | Prog Acc | Exe Acc | Prog Acc | |
| =1 | 76.30 | 75.66 | 70.27 | 68.77 | 73.70 | 71.25 | 4240 |
| =2 | 66.01 | 66.01 | 63.69 | 61.79 | 62.34 | 59.65 | 2300 |
| ≥3 | 31.78 | 31.10 | 31.65 | 31.65 | 28.57 | 23.80 | 594 |

Table 5 displays the models' performances when generating programs with different steps. ELASTIC (RoBERTa-large) performs better than FinQANet (RoBERTa-large)† when the program step is either 1 or 2, indicating ELASTIC also performs well on the shorter program steps. Surprisingly, with the program step increasing from 3, the accuracy for both ELASTIC and FinQANet tumbles by half compared with the performances on program steps equal 2. We suspect that the FinQA dataset lacks sufficient training examples for the data with more than 3 program steps. Table 5 shows that the number of training data with more than 2 program steps is 594 compared to the numbers of data available for program steps equal to 1 (4240) or 2 (2300). For a fair comparison, we retrained the FinQANet (RoBERTa-large) on the FinQA dataset, but ELASTIC still outperforms it in execution accuracy and program accuracy.

From Figure 2c, ELASTIC (RoBERTa-large) surpasses the FinQANet (RoBERTa-large) on MathQA dataset almost on every program step. Meanwhile, although MathQA is challenging, ELASTIC (RoBERTa-large) still achieves program accuracy over 80.0 when program steps are less than or equal to 8. The model's performance drops when the program steps are equal to 9 and 10, but starts to soar when the program steps are bigger than 10. This demonstrates that ELASTIC performs well when generating longer program steps. As shown in Figure 2c, we plot the accuracy for the operator generation, which ignores the correctness of operands generation. We could find that the operation accuracy is always higher than the program accuracy regarding different program steps (except for program steps equal to 12). This finding demonstrates the advantage of separating the generation

procedure for operators and operands. This finding also reveals that the wrong predictions are because ELASTIC selects the wrong operands. We suspect this is due to too much noise from the context.

Finally, our ELASTIC (RoBERTa-large) model (with approximately 500 million trainable parameters) outperforms Austin et al.'s smallest model [43] (with 8 billion trainable parameters) but only marginally underperforms their largest model (with 137 billion trainable parameters). The parameter size of these models are 1600 and 274000 times bigger than our model respectively, resulting the training resources required significantly considerable.

## 6 Conclusion and Future Work

This paper presents the numEricaL reASoning with adapTive symbolIc Compiler (ELASTIC) model aiming to solve the numerical reasoning over text problem. ELASTIC separates the generation of operators and operands, allowing the model to generate the long and complicated reasoning program. Also, ELASTIC is domain agnostic and supports diverse operators, increasing adaptability. In addition, we introduce the Memory Register and improve the performance of the model by using executable results from the preceding sub-programs. We evaluated the performance of the ELASTIC model on FinQA and MathQA datasets and conducted an extensive comparison with state-of-the-art models. The results show ELASTIC gained significant improvement over the state-of-the-art baselines. Furthermore, We investigated the model's performance in terms of different M-MDD, demonstrating the necessity of the Memory Register. Finally, we compared models' performances regarding different lengths of numerical reasoning programs, showing ELASTIC is adept at producing long numerical reasoning programs. In the future, we plan to improve the accuracy of matching numbers and entities of the text. In addition, ELASTIC requires annotated reasoning programs, which is labor intensive. It is worth investigating how to generate reasoning programs from the trained model.

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
