# ELASTIC: Numerical Reasoning with Adaptive Symbolic Compiler SUPPLEMENTARY MATERIAL

## 1 Appendix A: Training ELASTIC (RoBERTa-large) on Extended FinQA Dataset

One advantage of our ELASTIC model is that it is adaptable to the number of operands of an operator. We demonstrate this by evaluating ELASTIC on the MathQA dataset in the "Overall Results" section. However, another dataset we used for the evaluation, the FinQA dataset, only contains questions solved by operators with two operands. To test the advantage of our model on the FinQA dataset, we manually extend it by adding 30 and 20 questions for train and test data (named extended FinQA dataset), respectively (see Table 2 for one example of the extended questions). These questions are proposed based on the original passages in the FinQA dataset. In addition, they are about superlative questions, which require to be solved by using superlative operators (i.e., *smallest* and *biggest*). As a result, unlike questions from the original FinQA dataset, the numbers of operands used to solve these extended questions are not limited to two. Next, We trained ELASTIC (RoBERTa-large) on the train data from the combination of the extended FinQA dataset and the original FinQA dataset. Since the number of operands of an operator is not determined anymore, the Reasoning Manager of ELASTIC has to manage the Operands Generator to generate the correct number of operands in terms of the specific question. This increases the difficulty for the model to generate correct operands and makes the dataset more challenging. The results are shown in Table 1. For the performance on the combined test data (original FinQA + extended FinQA), ELASTIC (RoBERTa-large) achieves slightly lower scores (64.5 of Exec Acc and 63.8 of Prog Acc), compared to the results of ELASTIC (RoBERTa-large) achieved on original FinQA dataset (68.96 of Exec Acc and 65.21 of Prog Acc).[1] We also report the metric scores of ELASTIC (RoBERTa-large) achieved on test data from the extended FinQA dataset: 90.0 on both Exec Acc and Prog Acc. Note that the state-of-the-art model FinQANet cannot solve the extended FinQA dataset because it can only generate operators with two operands. These results show ELASTIC model solving questions that require the capability of generating operators with diverse numbers of operands.

Table 1: The performances of ELASTIC (RoBERTa-large) on the test data from the combination of the original FinQA dataset and extended FinQA dataset, and only on the test data from the extended FinQA dataset. Note that the model is trained on the train data from the combination of the original FinQA dataset and the extended FinQA dataset.

| Dataset (Test) | Exec Acc | Prog Acc |
|---|---|---|
| original FinQA + extended FinQA | 64.5 | 63.8 |
| extended FinQA | 90.0 | 90.0 |

Table 2 shows one example from the extended FinQA dataset. To solve this question, the model needs to select numbers relevant to the obligations of payments between 2007 and 2019, compare them, and select the biggest one. We observe that ELASTIC (RoBERTa-large) correctly selects

---

[1]See full results in "Overall Results" section.

Table 2: An example from the extended FinQA dataset. The "Prediction" refers to the generated numerical reasoning program from ELASTIC (RoBERTa-large). †: The passage is from the FinQA dataset, which is reorganized for better readability.

| Question | What is the biggest obligations of payments between 2007 and 2010? |
|----------|-------------------------------------------------------------------|
| Passage† | Contractual obligations and commercial commitments the following table (in thousands ):
The operating lease obligations of payments due by fiscal year total is $4819;
The operating lease obligations of payments due by fiscal year 2007 is $1703;
The operating lease obligations of payments due by fiscal year 2008 is $1371;
The operating lease obligations of payments due by fiscal year 2009 is $1035;
The operating lease obligations of payments due by fiscal year 2010 is $710;
The total obligations of payments due by fiscal year 2007 is $1903;
The total obligations of payments due by fiscal year 2008 is $1571;
The total obligations of payments due by fiscal year 2009 is $1235;
The total obligations of payments due by fiscal year 2010 is $710. |
| Prediction | *biggest*(1903, 1571, 1235, 710) |

the relevant numbers and applies the *biggest* operator to them. Worth mentioning, there are 9 numbers ($4819, 1703, 1371, 1035, 710, 1903, 1571, 1235, 710$) relevant to the "*payments*" in the question. ELASTIC only selects part of these numbers which are relevant to the "*obligations of payments*" asked by the question. This demonstrates that ELASTIC understands the aim of the question and solves it by generating the correct numerical reasoning program.

## 2 Appendix B: Comparison between Operation Length of the Golden Numerical Reasoning Program in MathQA and FinQA Datasets

Figure 1 compares the distribution of operation length of the golden numerical reasoning program in FinQA and MathQA datasets. We can see that the lengths of operation of most numerical reasoning programs in FinQA are between 2 and 4. In contrast, MathQA contains more golden numerical reasoning programs with operation lengths between 3 and 8. Obviously, MathQA contains longer numerical reasoning programs than FinQA. As a result, the MathQA dataset is more complicated than the FinQA dataset.

## 3 Appendix C: Case Study

Table 3 shows two cases of the predictions by ELASTIC (RoBERTa-large) on MathQA dataset.

- **Case 1** The question, in this case, requires to be solved by three operators with different numbers of operands, such as *sqrt* takes only one operand. We can see that ELASTIC (RoBERTa-large) generated the correct numerical reasoning program, by using constant *none* as the padding operand for the operators *sqrt* and *floor*. This case depicts one example when ELASTIC generates operators with different numbers of operands.

- **Case 2** This case shows the scenario where a long numerical reasoning program is used to solve the question. Although the operation length of the golden numerical reasoning program is 11 in this case, ELASTIC (RoBERTa-large) generates the correct program. Because ELASTIC separates the generation procedures for operators and operands, which prevents the potential interactive distraction between operators and operands. This makes ELASTIC less liable to being influenced by the cascading error.

## 4 Appendix D: Possible Explanation for Graph2Tree Poor Performance on FinQA Dataset

Section "Overall Results" shows that baseline Graph2Tree achieves only 0.37 accuracy on the FinQA test dataset, which is significantly lower compared to the accuracy of Graph2Tree on FinQA eval data (83.2 of program accuracy). We suspect this is due to the data leak problem existing in FinQA

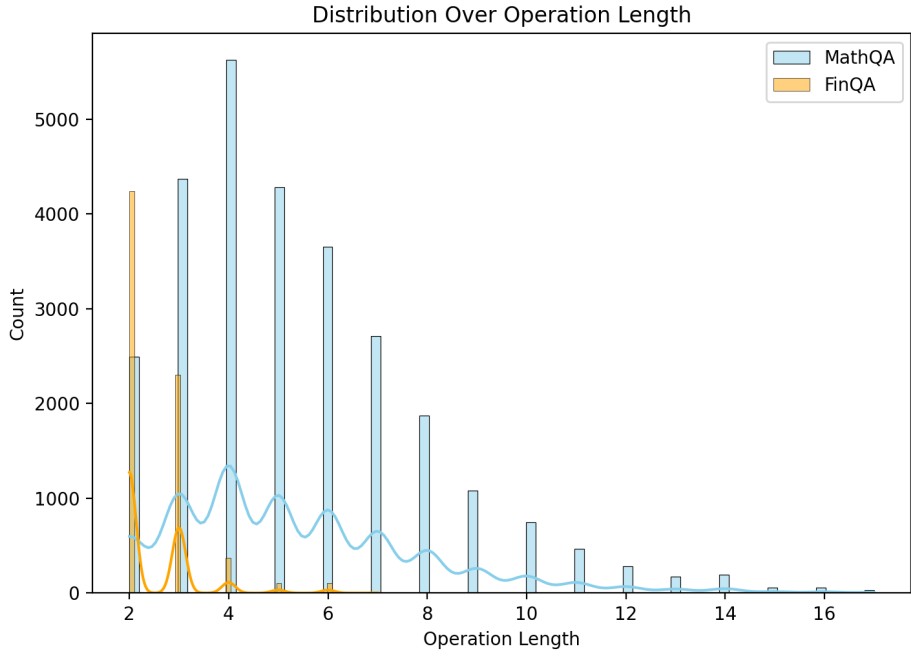

Figure 1: The distribution of Operation Length in FinQA and MathQA datasets.

Table 3: Two cases showing predicted reasoning program from the ELASTIC (RoBERTa-large). Except for the Prediction, the data are taken from the MathQA dataset.

| | | |
|---|---|---|
| Case 1 | Passage | if n is an integer and 101 n ^2 is less than or equal to 10000 , what is the greatest possible value of n ? |
| | Prediction | *divide*(n2, n0), *sqrt*(#0, none), *floor*(#1, none) |
| | Golden | *divide*(n2, n0), *sqrt*(#0), *floor*(#1) |
| Case 2 | Passage | Real - Estate salesman z is selling a house at a 25 percent discount from its retail price . Real - Estate salesman x vows to match this price and then offers an additional 5 percent discount. Real - Estate salesman y decides to average the prices of salesmen z and x then offer an additional 30 percent discount. Salesman y's final price is what fraction of salesman x's final price. |
| | Prediction | *subtract*(const_100,n1), *subtract*(const_100,n0), *subtract*(const_100,n2), *divide*(#0, const_100), *multiply*(#3,#1), *add*(#4,#1), *divide*(#5, const_2), *multiply*(#6,#2), *divide*(#7,const_100), *divide*(#8,#4), *multiply*(#9, const_10) |
| | Golden | *subtract*(const_100,n1), *subtract*(const_100,n0), *subtract*(const_100,n2), *divide*(#0, const_100), *multiply*(#3,#1), *add*(#4,#1), *divide*(#5, const_2), *multiply*(#6,#2), *divide*(#7,const_100), *divide*(#8,#4), *multiply*(#9, const_10) |

train and eval data. Specifically, Unlike ELASTIC and other baselines that generate the indices of the operands in the passage, Graph2Tree generates the operand values directly. As a result, Graph2Tree defines the decoding vocabulary containing all operands in the train and eval data. This reveals that Graph2Tree cannot produce undefined operands, which are widely existed in the test data. To demonstrate our suspicion, we calculate the overlap of occurring operands in the FinQA dataset, between train and eval data, and train data and test data. The original code of Graph2Tree creates the decoding vocabulary by all operands that occur in both train and eval datasets, so that the overlap rate is 100% between train and eval data. In contrast, the overlap rate between train and test data is only 56%. In other words, Graph2Tree cannot generate numerical reasoning programs which contain the left 44% unseen operands, where the proportion of these numerical reasoning programs in the FinQA test dataset is 33%. The number indicates that Graph2Tree could achieve at most 33% program accuracy on the FinQA test data.

# 5  Appendix E: An Example to Illustrate Notations in Table 2

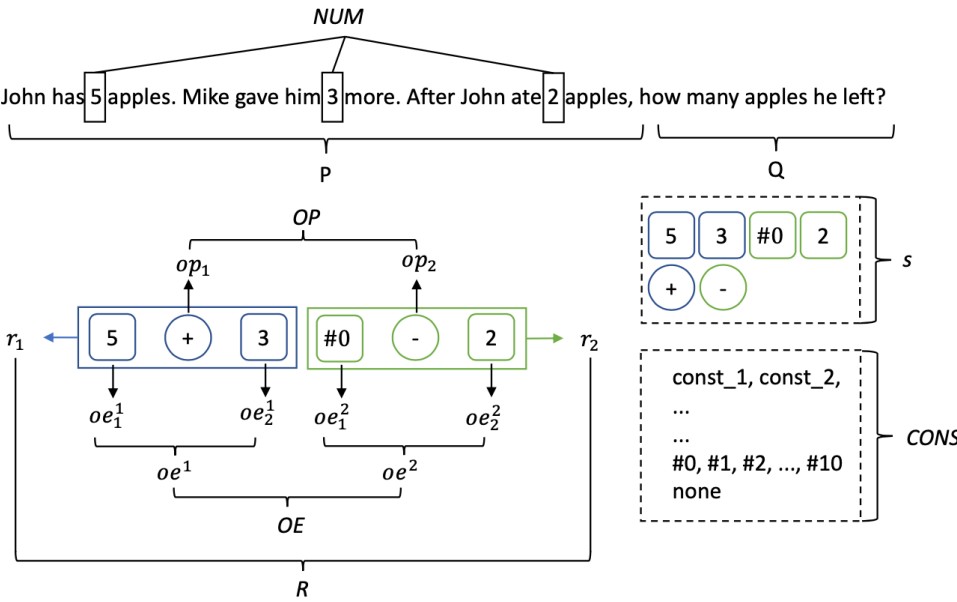

Figure 2: An example of math word problems, using nations defined in Table 2.

Figure 2 shows an example of a math problem and its numerical reasoning programs. The problem text $P$ and question text $Q$ are combined, which contains three numbers: 5, 3, 2, denoted by *NUM*.

The reasoning program $R$ contains two sub-programs, "$5 + 3$" (denoted as $r_1$) and "$\#0 - 2$" (denoted as $r_2$). The first sub-program $r_1$ contains one operator "$+$" (denoted as $op_1$) and two operands "5" (denoted as $oe_1^1$) and "3" (denoted as $oe_2^1$). The second sub-program $r_2$ contains one operator "$-$" (denoted as $op_2$) and two operands "$\#0$" (denoted as $oe_1^2$) and "2" (denoted as $oe_2^2$). "$\#0$" is pre-defined in the constant vocabulary, which is denoted as *CONS*.

The $op_1$ and $op_2$ are belong to all mathematical operators *OP*. The $oe_1^1$, $oe_2^1$, $oe_1^2$, and $oe_2^2$ belong to all operands *OE*. We regard *OP*, *OE* as symbols $s$.

# 6  Appendix F: An Example to Show How Separated Modules Work

Figure 3 shows the generation process of equations: "$5 + 3 - 2$", which is represented as two sub-programs: "$+, 5, 3$" and "$-, \#0, 2$". At the beginning, the Operator Generator produces "$+$" by sampling from the OP decoding vocabulary (refer to Equation (6)). Next, the generation for the operator is suspended, and the Reasoning Manager guides Operands Generator to produce "3" and

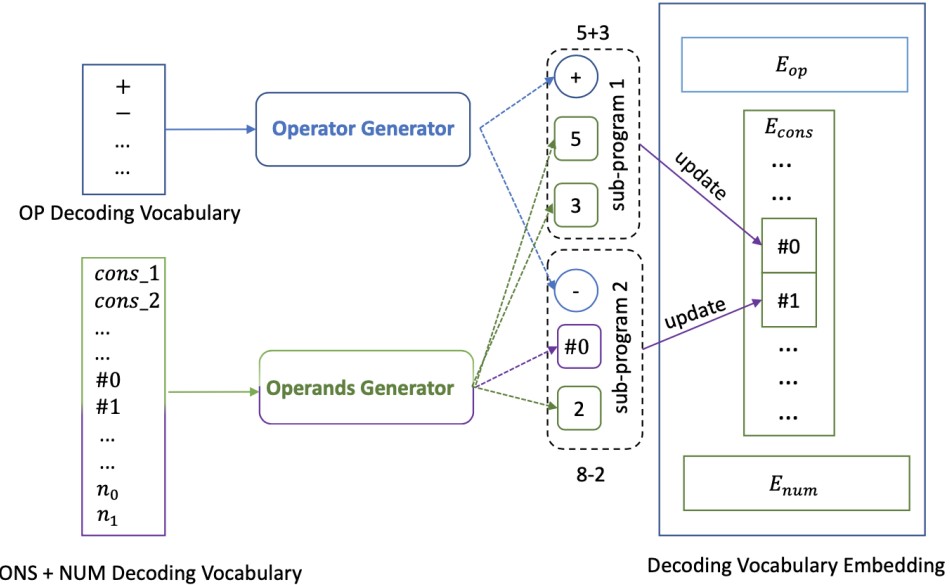

Figure 3: An Example to Show How Separated Modules Work

"2" sampling from the NUM Decoding Vocabulary (refer to Equation (7)). After the first sub-program is complete, the Memory Register replaces the first cache token "#0" embedding with the guidance vector from Reasoning Manager. When generating the second sub-program, the Reasoning Manager enables the Operator Generator to produce the operator "−" for the second sub-program. Since the first operand in equation "8 − 2" refers to the executable result from the first sub-program. As a result, the Operands Generator produces "#0", which refers to the executable result of the first sub-program. Finally, after operand "2" is generated, the generation for the second sub-program completes. Likewise, the Memory Register updates the second cache token "(#1)" embedding with the guidance vector.