# OpenReview forum: "ELASTIC: Numerical Reasoning with Adaptive Symbolic Compiler"
_NeurIPS.cc/2022/Conference — NeurIPS 2022 Accept_

### Official Review · Reviewer_2nDD · 2022-07-10

**Rating:** 6
**Confidence:** 3
**Soundness:** 3 good
**Presentation:** 3 good
**Contribution:** 3 good

**Summary:**

This paper proposes a new model called numEricaL reASoning with adapTive symbolIc Compiler (ELASTIC) to solve numerical reasoning QA problems. ELASTIC separates the generation of operators and operands, and is adaptable to the number of operands, making it more accurate and flexible. They show promising results on two datasets including FinQA and MathQA.

**Questions:**

1. Comparison with large language models. The authors claim that pretrained language models fall short of numerical reasoning from the text. However, recent advances in program synthesis show that large language models perform well in this area: they achieve 83.8% on MathQA which outperforms the 83.00% in this paper. I agree that this paper is valuable but for the completeness could you make more comparisons with these large language models in terms of number of parameters, runtime costs, and generalization ability, etc.?

2. Some details introduced in Section 3 are unclear. For example, what does “NGS” mean? What’s the size of the memory register (size = max program steps or size = 1)? Is it possible to train another network to predict which memory slot should be overwritten when the memory is full [2]?

3. Although the authors say ELASTIC is domain agnostic, it seems that the performance of these program-based models can be significantly influenced by the programming language. Can ELASTIC solve MathQA-python[3], which replaces the original MathQA DSL with python?

[1] Program Synthesis with Large Language Models

[2] Automatic Program Synthesis of Long Programs with a Learned Garbage Collector

[3] https://github.com/google/trax/blob/master/trax/examples/MathQA_Python_generation_notebook.ipynb

**Limitations:**

Please see questions listed above.

**Strengths And Weaknesses:**

Numerical reasoning is an important and challenging task of AI. Although the idea of generating operators and operands separately is already been adopted in other areas like program synthesis, it is still novel and valuable in numerical reasoning to alleviate the cascading error problem. Experimental results are solid and promising, which well supports the authors’ claims.

The writing needs to be improved, especially the method part, which is somewhat confusing. It seems that the method proposed cannot be applied to QA problems without labeled programs. Also, the influence of different programming language has not been discussed.

---

> ### Author Response · Authors · 2022-08-02
> **Author response to R_2nDD**
>
> We thank the R_2nDD for providing two references and one link. We uploaded a revised paper and supplementary material version, highlighting changed parts in blue. In addition, We address the comments as follows.
>
> - R_2nDD_c1: “The writing needs to be improved … which is somewhat confusing.”
>    - We rewrote the introductory part of Section 3 and provided an example in Appendix F to show how different modules of ELASTIC work. In addition, the added Figure 3 in Appendix F describes how the Memory Register stores and uses the cache token. To make Table 2 clear, we added Appendix E to illustrate the notations of Table 2 using an example. Finally, we described the limitation of  ELASTIC in terms of more data required in the conclusion section. The modifications could be observed in the revised version of the paper and supplementary material, which are uploaded for the rebuttal phase.
>
> - R_2nDD_c2: “Is seems that the method proposed … has not been discussed.”
> - R_2nDD_c3: “Although the authors say ELASTIC is domain agnostic … MathQA DSL with python?”
>    - Thanks for providing the paper and code. We decide to answer the above questions in one block.
>    - The domain-agnostic refers to our model’s generalizability to different data from different domains. For example, FinQA is from the finance domain, and MathQA is from the math domain. Both datasets require numerical reasoning abilities but involve different operators and operands. Table 3 shows that ELASTIC achieves SOTA results on two datasets: FinQA and MathQA.
>    - We have looked at the data from MathQA-python, and found the difference between original MathQA with MathQA-python is that they use an alternative way to represent the intermediate reasoning process. As a result, ELASTIC could solve MathQA-python by adopting several prerequisite steps.  Take one example from the MathQA-python, which represents the intermediate reasoning process as “n0 = 5, n1 = 31.1, t0 = n1 + 100.0, t1 = 100 – n0, t2 = t0 * 100, t3 = t2 / t1, answer = t3 -100”. To make ELASTIC generate this, we should create a new DSL for MathQA-python. For example, replacing the “add” from DSL defined in the paper with “+”. Next, we should discard the previous two equations, where the values of n0 and n1 have been extracted by ELASTIC during the pre-processing step. Consequently, the intermediate reasoning process of this example from the MathQA-python is changed to “+(n1, 100), -(const_100 - #0), *(#0, 100), /(#2, #1)”, which could be generated by ELASTIC directly. As a result, the generation processes for MathQA and MathQA-python by ELASTIC are the same, and the only difference between generation processes is how to define the operators and operands in DSL.
>
> - R_2nDD_c4: “Comparison with large language models … and generalization ability, etc.?”
>    - Recent large language models show promising results in numerical reasoning tasks. However, the parameter size of the smallest model in Table 6 from paper [1] provided by R_2nDD is 8B, which is 1600 times larger than our biggest model ELASTIC-RoBERTa-large (approximately 500M in parameter size). Despite this vast gap between the model size, ELASTIC-RoBERTa-large still outperforms the 8B model in paper [1].
>    - It is true that the large language models in the size of 68B/137B in paper [1] achieved 2.8%/3.8% incremental accuracies compared to ELASTIC-RoBERTa-large on MathQA dataset. However, ELASTIC-RoBERTa-large could be trained on a single A100 GPU, which hugely saves computational cost and time.
>
> - R_2nDD_c5: “Some details introduces in Section 3 … when the memory is full [2]?"
>    - We have read paper [2], particularly on why they train a network to predict whether the variable need to be dropped. The reason for [2] introducing this network is that they have a fixed maximum number of operands for an operator. For ELASTIC, such a limitation doesn’t exist. The Operands Generator of ELASTIC stops when producing a “none” token after an operator, and the numbers of operands following different operators in one reasoning program could be different. Thus, we do not need to discard the cache token since we will not meet the situation of full memory. For example, the cache token #n refers to the n-th previous sub-program’s result. And #n is just like an operand with its own embedding vector. After each sub-program is generated, the embedding vector of #n is updated by the Memory Register. As a result, their embeddings are not static. For example, given to programs “add(3,2),subtract(#0,1)” and “exp(x), add(1,#0)”, both #0s refer to the first sub-program of each reasoning programs, however, #0 in different programs refers to different sub-program. In this example, it refers to either “add(3,2)” in the first program example, or “exp(x)”in the second program example. Since the embedding vector of #0 is replaced by the information of the sub-program once the first sub-program is generated, the embedding vector of #0 is unique in different reasoning programs.

---

### Official Review · Reviewer_vH6J · 2022-07-11

**Rating:** 5
**Confidence:** 4
**Soundness:** 3 good
**Presentation:** 2 fair
**Contribution:** 2 fair

**Summary:**

The authors present a novel model (ELASTIC) for numerical reasoning problems. The proposed model consists of an encoder and a compiler which separates the generation of operators and operands. The authors claim that the separated generators enhance the adaptability and elasticity of the model. The proposed model also incorporates a memory register to improve the model’s performance on the numerical problems with large M-MDD. The proposed model achieves state-of-the-art performance on two datasets: FinQA and MathQA.


**Questions:**

Results in Figure 2 are a bit confusing. The accuracy with M-MDD=2 is much lower than that with M-MDD=3 in figure 2 (a), why? What’s the possible reason of the large fluctuations in figure 2 (b) and 2(c)? Why the performance of ELASTIC starts to soar when the program steps are bigger than 10 in figure 2 (c)?  Does the similar phenomenon exist in the other dataset?


**Limitations:**

The experiments might be designed more carefully. When analyzing the necessity of memory register, the effect of separated generators is not disentangled with the effect of the memory register. For more effective analysis, there should be a setting ELASTIC w/o separated generator, or FinQANet w memory register if possible.

To make section 3 easier to follow, it might be better to use more textual examples instead of notations when possible.

Typos: line 107-108, 144, 262


**Strengths And Weaknesses:**

Strengths: Numerical reasoning is an important area in machine learning and natural language processing. This work proposes novel methods to enhance the numerical reasoning ability of the model. On one hand, the generation of operators and operands are separated. On the other hand, a memory register is introduced to make it easier for the model to use the previous sub-program results. The proposed model outperforms FinQANet on both FinQA and MathQA.

Weaknesses:
1. The motivation and effectiveness of separating the generation of operands and operators is not clear. The authors claim that previous work that does not use separated generators suffers from cascading errors when encountering complicated tasks (line 36-37), but the results shown in figure 2 (c) and Table 4 may not be sufficient to support this claim. The proposed model does not consistently stabilize the performance on long reasoning programs.

2. The improvement achieved by memory register seems to be marginal (Table 4).

3. The writing should be improved. The approach section is not easy to follow. The authors use notations throughout this section and some terms and notations are not well explained at their first appearance.

For example, DSL (line 108), NGS (line 136), W_{op} in eq 6, and the index i in eq 6 and 7 have different meanings.

Figure 1 shows the reasoning manager generates g^{op} and g^{oe} in two different flows and the inputs are different, but the difference is not explained in line 134-143. Although the following introduction of two generators explains the difference, it might be better to move this content to line 134-143. Figure 1 and line 160-172 shows how the results from the RM updates the memory register but does not show how the vectors stored in the memory register is used during the generation.  I guess there should be another arrow from the memory register to the generator in Figure 1.  Please clarify.

---

> ### Author Response · Authors · 2022-08-02
> **Author response to R_vH6J**
>
> We thank the R_vH6J for the constructive comments. We uploaded a revised paper and supplementary material version, highlighting changed parts in blue. In addition, we provide some specific responses and clarifications as follows.
>
> - R_vH6J_c1: “The motivation and effectiveness … long reasoning programs.”
>    - Separating the operators generation and operands generation allows ELASTIC to support diverse operators with any number of operands. Methods without separating the generations, like FinQANet, can only support a fixed number of operands following an operator,  and are usually hardcoded in the implementation. One example of such a hardcoding can be observed from the FinQANet code: https://github.com/czyssrs/FinQA/blob/main/code/generator/Model_new.py#L277.
>    - Figure 2(c) shows ELASTIC achieves higher score if only considering the accuracy of generated operators (orange line), compared to the considering both operators and operands (blue line). This indicates that the wrong generated operands could result in the wrong generated program. However, ELASTIC could escape from the cascading error from the wrong generated operands, and generates the correct operators.
>    - Table 4 is not used to demonstrate the effectiveness of separating the operators generation and operands generation. However, Figure 2 compares ELASTIC with FinQANet (which does not separate the generation processes) and demonstrates the improvements brought by the separate modules design of ELASTIC.
>
> - R_vH6J_c2: “The improvements achieved by memory register seems to be marginal (Table 4).”
> - R_vH6J_c3: “The experiments might be designed … if possible.”
>    - The most significant advantage of ELASTIC is the separation of the operators generation and operands generation, which is the integral and core component of the model. If we remove this component from ELASTIC, it will lose the advantage of being adaptable to the number of operands following an operator. However, we compare ELASTIC with FinQANet, the model that doesn’t separate the generators. As shown in Table 3, ELASTIC outperforms FinQANet.
>    - Also, FinQANet is with Memory Register by nature (we mentioned this in lines 256-257 of the original paper). Therefore, we have compared with FinQANet with Memory Register as R_vH6J requested. Table 2(a)(b) compare ELASTIC (w Memory Register) with FinQANet (w Memory Register), the better results achieved by ELASTIC (w Memory Register) shows the superiority of our Memory Register design. Finally, Table 4 points out the necessity for the Memory Register, by showing ELASTIC (w Memory Register) performs better than ELASTIC (wo Memory Register)
>    - The reason for the marginal improvement shown in Table 4 is that it compares models’ performances on the entire test data, where some examples contain reasoning programs without using the results from the previous sub-programs. The proportions of these programs are 57% and 10% in FinQA and MathQA datasets, respectively.
>
> - R_vH6J_c4: “The wring should be improved … Please clarify.”
> - R_vH6J_c5: “To make section 3 easier to follow … when possible.”
> - R_vH6J_c6: “Typos: line 107-108, 144, 262.”
>    - We apologize for the typos, and sincerely appreciate for pointing them out.
>    - As suggested, we added one sentence to explain the different inputs to Reasoning Manager. Also, we rewrote the introductory part of Section 3 and provided an example in Appendix F to show how different modules of ELASTIC work. In addition, the added Figure 3 in Appendix F describes how the Memory Register stores and uses the cache token. To make Table 2 clear, we added Appendix E to illustrate the notations of Table 2 using an example. The modifications could be observed in the revised version of the paper and supplementary material, which are uploaded for the rebuttal phase.
>
> - R_vH6J_c7: “Results in Figure 2 are a bit confusing … M-MDD=3 in figure 2 (a), why?”
>    - Figure 2(a) shows that both ELASTIC and FinQANet encounter this problem. This is because the small size of testing data causes a large variance. In particular, the number of examples in FinQA dataset with M-MDD equal to 2 and 3 in the test data is only 20 and 13, respectively, compared to the numbers of examples with M-MDD equal to 0 and 1 are 657 and 456, respectively.
>
> - R_vH6J_c8: “What’s the possible reason … exist in the other dataset?”
>    - MathQA test data contains examples with the M-MDD of 7,8,9,10 in the size of 28,7,12,7. The small size of the examples leads to fluctuations (or high variance), which could be observed from both ELASTIC and FinQANet. The reason for the performance of ELASTIC starts to increase when the program step becomes bigger than 10 is also due to that the test examples with lengths of program steps bigger than 10 are limited in size of 64, 38, 29.

---

> > ### Comment · Reviewer_vH6J · 2022-08-07
> > **The revised version has addressed most of the questions**
> >
> > The writing has been improved in the revised version and most of the questions have been resolved.  Although the authors' response still does not convince me of an excellent technical novelty, it's reasonable to raise the overall rating (borderline accept).

---

### Official Review · Reviewer_3k7j · 2022-07-11

**Rating:** 8
**Confidence:** 4
**Soundness:** 4 excellent
**Presentation:** 3 good
**Contribution:** 3 good

**Summary:**

The authors propose ELASTIC, a multi-modular network for numerical reasoning on MathQA and FinQA datasets. It consists of an encoder and a compiler with four different modules. Each module is specifically designed to solve a part of the numerical reasoning task. For example, the operator generator decides the next operation, and then the operand generator picks the operands to be used. The memory register stores some intermediate computations for future use. Overall, they achieve strong improvements over prior baselines. Additionally, this method generalizes better to higher program steps.

**Questions:**

Writing suggestions:
* Please rewrite Section 3 introduction using less technically repetitive terms and more general ones/synonyms. Currently, it's a bit hard to read through.
* Please add a figure demonstrating the terms in Table 2 clearly.
* Similarly, I feel it would help to have a motivating figure to explain the need and functionality of the memory register, for completeness.

**Limitations:**

Please try to add a section specific to the limitations of the method. Other than the generalization issues, I feel these modular approaches might need more training data compared to the end-to-end frameworks. You may consider discussing some points along these lines.

**Strengths And Weaknesses:**

Strengths:
* The paper is very nicely motivated, with good supporting experiments and ablations
* The design of each component is generally well described and easy to follow through (except some parts mentioned below)
* This direction of using specific modules for a fixed set of tasks (similar to NMN's), is indeed very interesting for this area

Weakness:
* Why is NumNet+ the only baseline from the DROP dataset leaderboard used in the paper? There are several other meaningful baselines that can be tried by just keeping the arithmetic heads at classification time. Please try adding more of these baselines, as applicable, to make Table 3 stronger.

---

> ### Author Response · Authors · 2022-08-02
> **Author response to  R_3k7j**
>
> We thank the reviewer R_3k7j for the positive and constructive feedback. We uploaded a revised paper and supplementary material version, highlighting changed parts in blue. In addition, we address the comments as follows.
>
> - R_3k7j_c1: “Why is NumNet+ the only … Table 3 stronger.”
>    - Besides NumNet/NumNet+, NeRd (in Table 3) is one of the strong baselines for DROP dataset. Paper [1] evaluated NeRd on both DROP and MathQA datasets. The reasons for us to select NeRd and NumNet/NumNet+ from the DROP leaderboard are as follows:
>      1. We couldn’t evaluate our model against other models in the DROP leaderboard that their codes, papers, or both are not provided. Here is a list showing the status of codes and papers for models from the DROP leaderboard, which achieve accuracies higher than 0.8: OPERA(no paper, no code), QDGAT(has paper, no code), sna_albert + Ensember(no paper, no code), na_albert+(no paper, no code), ALBERT-Calculator(based on [2], both it and [2] have no code).
>      2. We compared  against NeRd which adopts a different method than NumNet/NumNet+. NeRd represents the reasoning program as the symbolic nested program, whereas NumNet/NumNet+ assign the “+/-” signs for numbers, which is turned into a classification problem. As a result, we think that they are two strong baselines.
>      3. Other models such as, BERT-Calculator [2], MTMSN, NAQANet+, are considered similar to NumNet/NumNet+ but with lower performances. Particularly, NumNet/NumNet+ adopt the same arithmetic head for numerical question types as them. Therefore, we decide not to use them as baselines.
>
> - R_3k7j_c2: “Please rewrite Section 3 … to read through.”
> - R_3k7j_c3: “Similarly, I feel it … for completeness.”
> - R_3k7j_c4: “Please add a figure … Table 2 clearly.”
> - R_3k7j_c5: “Please try to add a section … along these lines.”
>    - We decide to answer the above four comments in one block. We rewrote the introductory part of Section 3 and provided an example in Appendix F to show how different modules of ELASTIC work. In addition, the added Figure 3 in Appendix F describes how the Memory Register stores and uses the cache token. To make Table 2 clear, we added Appendix E to illustrate the notations of Table 2 using an example. Finally, we described the limitation of ELASTIC in terms of more data required in the conclusion section. The modifications could be observed in the revised version of the paper and supplementary material, which are uploaded for the rebuttal phase.
>
> - References:
>    - [1] Neural Symbolic Reader: Scalable Integration of Distributed and Symbolic Representations for Reading Comprehension
>    - [2] Giving BERT a Calculator: Finding Operations and Arguments with Reading Comprehension

---

### Meta-Review · Area_Chair_NeKg · 2022-08-27

**Recommendation:** Accept
**Confidence:** Certain

**Metareview:**

The paper presents a new model called Numerical Reasoning with Adaptive Symbolic Compiler that is able to perform numerical reasoning tasks. One of the key ideas in this model is to separate the generation of operators and operands and to include a memory register to remember intermediate values. The method is compared against FinQANet and shows good results on both the MathQA and FinQA datasets. In the reviews and rebuttal, it emerged that the performance of this model is marginally lower than the performance of state-of-the-art large language models on this task; however, the difference in size and training resources required by this model compared with the LLM is so vast that this still represents a significant advance over the state of the art. It would nevertheless be good for this to be mentioned in the paper itself. Overall, this is a novel approach that advances an important problem.

**Award:**

No

---

### Decision · Program_Chairs · 2022-09-14

Accept